# Epidemiological Investigation and Genetic Analysis of Pseudorabies Virus in Yunnan Province of China from 2017 to 2021

**DOI:** 10.3390/v14050895

**Published:** 2022-04-25

**Authors:** Jun Yao, Juan Li, Lin Gao, Yuwen He, Jiarui Xie, Pei Zhu, Ying Zhang, Xue Zhang, Luoyan Duan, Shibiao Yang, Chunlian Song, Xianghua Shu

**Affiliations:** 1Yunnan Tropical and Subtropical Animal Virus Diseases Laboratory, Yunnan Animal Science & Veterinary Institute, Kunming 650224, China; yaojun_joshua@hotmail.com (J.Y.); 13987123280@163.com (L.G.); heyuwen1117@163.com (Y.H.); xjr1990123@sina.com (J.X.); zpcau@sina.com (P.Z.); yangsb3799@sina.com (S.Y.); 2Yunnan Sino-Science Gene Technology Co., Ltd., Kunming 650501, China; ynndlj@126.com (J.L.); xuezhangx@outlook.com (X.Z.); 3College of Animal Medicine, Yunnan Agricultural University, Kunming 650201, China; ynau_zhangying@163.com; 4College of Veterinary Medicine, China Agricultural University, Beijing 100193, China; sy20193050805@cau.edu.cn

**Keywords:** pseudorabies virus, seroprevalence, epidemiology, phylogenetic analysis, variants

## Abstract

In recent years, the prevalence of pseudorabies virus (PRV) has caused huge economic losses to the Chinese pig industry. Meanwhile, PRV infection in humans also sounded the alarm about its cross-species transmission from pigs to humans. To study the regional PRV epidemic, serological and epidemiological investigations of PRV in pig populations from Yunnan Province during 2017–2021 were performed. The results showed that 31.37% (6324/20,158, 95% CI 30.73–32.01) of serum samples were positive for PRV glycoprotein E (gE)-specific antibodies via enzyme-linked immunosorbent assay (ELISA). The risk factors, including the breeding scale and development stage, were significantly associated with PRV seroprevalence among pigs in Yunnan Province. Of the 416 tissue samples collected from PRV-suspected pigs in Yunnan Province, 43 (10.33%, 95% CI 7.41–13.26) samples were positive for PRV-*gE* nucleic acid in which 15 novel PRV strains from these PRV-positive samples were isolated, whose *gC* and *gE* sequences were analyzed. Phylogenetic analysis showed that all 15 isolates obtained in this study belonged to the genotype II. Additionally, the *gC* gene of one isolate (YuN-YL-2017) was genetically closer to variant PRV strains compared with others, while the *gE* gene was in the same clade with other classical PRV strains, indicating that this isolate might be a recombinant strain generated from the classical and variant strains. The results revealed the severe PRV epidemic in Yunnan Province and indicated that PRV variants are the major genotypes threatening the pig industry development.

## 1. Introduction

Pseudorabies (PR) is a devastating infectious disease that poses a huge threat to the development of the pig industry worldwide [1]. The causative agent of PR, pseudorabies virus (PRV) or Suid herpesvirus (SuHV-1), is an enveloped double-stranded DNA virus that belongs to the subfamily *Alphaherpesvirinae* of the family of *Herpesviridae* [2]. Pigs are known as the natural host and reservoir for PRV. The clinical symptoms of pigs infected with PRV vary depending on the growth stages: in newborn piglets, PRV infection causes severe diarrhea, vomiting, and neurological symptoms, resulting in high morbidity; in pregnant sows, PRV infection leads to reproductive failure [2,3]. Moreover, PRV has an intensive cross-species transmission capacity, which can infect a wide variety of animals, such as pigs, ruminants, carnivores, bears, etc. [4]. Notably, PRV transmission from pigs to humans has raised worldwide concerns since Chinese researchers recently have successfully isolated a variant PRV strain from an acute human encephalitis case [5].

Since the first detection of PRV in the United States, the disease caused by this pathogen has been observed in many countries, including Canada, China, and Hungary [6]. PR has been successfully controlled or eradicated in some countries or regions, such as Canada and Mexico, due to the application of multiple diagnosis approaches and glycoprotein E (gE)-deleted live or attenuated PRV vaccines [2]. However, this infectious disease remains widely prevalent in Chinese populations. Since late 2011 especially, PRs caused by PRV variants have frequently erupted in some Bartha-K61-immuized pig farms in China [7,8]. Subsequent experiments showed that the Bartha-K61 vaccine could not provide complete protection against these variants [8].

Currently, PRV strains are composed of two genotypes (genotype I and genotype II). PRV strains from Europe and USA belong to the genotype I, while most of genotype II PRV strains are isolated from Asian countries, mainly in China [2]. Moreover, the genotype II strains can be further divided into two sub-genotypes (classical PRV strains and variant PRV strains) [2]. According to the genetic characteristics among different PRV genotype strains, several amino acid (aa) insertions and deletions were observed, for example, the PRV genotype II strains have a 3-aa continuous deletion (^75^VPG^79^) in the UL27 gene and a 7-aa continuous insertion (^63^AASTPAA^69^) in the UL44 gene compared with PRV genotype I strains [9].

An investigation of the prevalence of PRV is required to build up strategies to control and even eradicate PR and minimize the risk of humans contacting this infectious pathogen. Though the prevalence and genetic characteristics of PRV have been documented in several regions or provinces of China [2,3,10,11], the relevant information in Yunnan Province in recent years is still not available. To fill in this gap, 20,158 pig serum samples were collected from 2017 to 2021 to investigate the epidemiology of PRV in Yunnan Province. Furthermore, the genetic characteristics of 15 newly isolated PRV strains were analyzed based on their *gC* and *gE* sequences.

## 2. Materials and Methods

### 2.1. Samples Collection

A total of 20,158 pig serum specimens were collected from 573 pig farms between March 2017 and December 2021, which nearly covered the entire Yunnan Province, China. The sampled pigs were chosen according to the breeding scale and breeding model. In brief, approximately equal numbers of specimens were collected from different growth stages (sucking piglets, nursery pigs, fattening pigs, sows, and gilts). Meanwhile, approximately equal sampling frequency was applied; 10, 25~30, and 50~60 serum samples were collected from each small (<100 sows), medium (100~500 sows), and large-scaled pig farm (>500 sows), respectively. In addition, tissue samples (such as brain, lymph node, lung, and kidney) were collected from 416 PRV infection-suspected pigs in 107 farms; the clinical symptoms of these diseased pigs mainly included encephalitis, diarrhea, fever, etc. The specimens were collected with standard procedures and delivered to Yunnan Animal Science and Veterinary Institute in a cold environment. Detailed information of each sample was documented.

### 2.2. Serological Detection of Anti PRV-gE Antibodies

Anti-gE antibodies in each serum sample were detected with Pseudorabies Virus (PRV)-gE antibody ELISA Kits (Cat: CP144, IDEXX Laboratories, Westrook, ME, USA) following the manufacturer’s instructions, which could be used to differentiate the vaccine strain or field strain-infected pigs.

### 2.3. Virus Detection and Isolation

Viral DNA were extracted from the tissue samples using a DNA Isolation Kit (Genenode Biotech Co.Ltd., Beijing, China) according to the manufacturer’s instructions. PCR was performed targeting the partial PRV-*gE* gene, with primers gE-F/R (gE-F: 5′-CCCAACGACACGGGCCTCTA-3′; gE-R:5′-GCACAGCACGCAGAGCCAGA-3′). The virus was isolated from PRV-positive tissue samples for subsequent experiments. Briefly, the tissue samples were homogenized and subjected to three freeze–thaw cycles. The supernatants, containing PRV virus, were filtered through a 0.22 µm filter after centrifugation and inoculated into a monolayer of BHK-21 or ST cells, which were cultured in a 5% CO_2_ incubator at 37 °C. The supernatants and cells with obvious cytopathic effects (CPE) were harvested for plaque purification assays [3] and molecular identification by real-time PCR assays. Viral titers were determined by the Reed–Muench method in ST cells and the 50% lethal dose (LD_50_) of which in mice models were calculated as described by Luo et al. [12].

### 2.4. Sequencing and Genetic Analysis

PCR was performed to amplify the complete sequences of *gE* and *gC* of 15 novel PRV strains as described previously [2]. The positive PCR products were purified and cloned into the pUCm-T vector. The plasmid carrying either the *gE* or *gC* gene was sequenced in duplicate. The full-length of *gE* or *gC* sequences of 15 newly isolated PRV strains and reference strains were compared using the DNAStar version 7.10 software. The phylogenetic tree based on the *gE* or *gC* gene was generated using the neighbor-joining (NJ) method in MEGA X software, with 1000 bootstrap replicates [13]. Detailed information of 15 novel PRV isolates and reference strains were available in the NCBI database as shown in Table 1.

### 2.5. Data Analyses

The seroprevalence of PRV in pigs was presented as the minimum infection rate (MIR) with 95% confidence intervals (CIs). The statistical significance of PRV-gE seroprevalence among different groups was analyzed using a Chi-square test in SPSS 21.0 software (SPSS Inc., Chicago, IL, USA). A difference with a *p*-value < 0.05 was considered statistically significant.

## 3. Results

### 3.1. Seroprevalence of PRV-gE in Yunnan Province during 2017–2021

In total, 573 pig farms were included in this survey, where nearly all sampled pigs had been immunized with an attenuated PRV vaccine (Bartha-K61 or HB-98 strain) or inactivated PRV vaccine. Of the collected serum samples, 6324 out of 20,158 samples were seropositive for PRV-gE specific antibodies, contributing to the overall positive rate of 31.37% (95% CI 30.73–32.01). The seroprevalence rates of PRV-gE from March 2017 to Augest 2018, September 2018 to January 2020, and April 2020 to December 2021 were 29.25% (2355/8051), 41.48% (2449/5904), and 24.50% (1520/6203), respectively (Table 2) (*p* < 0.01).

In terms of pig herds, the average PRV-gE seroprevalence rate in piglets (16.72%, 442/2644) was significantly lower than these of other development stages of pigs (25.29~40.94%) (*p* < 0.01) (Table 2). Moreover, we further investigated the seroprevalence of PRV in pig farms with different breeding scales, which showed that the lowest seroprevalence was observed in medium scale farms (24.80%, 1556/6273), followed by small-scale farms and large-scale farms at 33.45% (3495/10,447) and 37.03% (1273/3438), respectively (*p* < 0.01) (Table 2).

### 3.2. PRV Detection and Viral Isolation

As shown in Table 3, of the 416 tissue samples collected from PR-suspected pigs, 43 (10.33%, 95% CI 7.41–13.26) samples were positive for PRV-gE nucleic acids. The detection rate of PRV among collected samples from March 2017 to August 2018, September 2018 to January 2020, and April 2020 to December 2021 were 9.04% (16/177), 14.56% (15/103), and 8.82% (12/136), respectively (*p* > 0.05). In terms of tissue samples from the pigs with different clinical symptoms, the positive rates of PRV infection among aborted fetuses (13.89%, 15/108) and piglets with neurological symptoms (18.07%, 15/83) were higher than other samples (5.78%, 13/225) (*p* < 0.01).

To further investigate the genetic features of PRV strains prevalent in Yunnan Province in recent years, 15 PRV strains were successfully isolated from the PRV-positive samples, purified via plaque purification, and further validated by PCR. The viral titers of these PRV strains were determined via the Reed–Muench method in ST cells, varying from ~10^5.25^ to 10^7.4^ TCID_50_/0.1 mL (Table 1). The subsequent animal experiments showed that the LD_50_ of 15 novel PRV strains to six-week-old female Kunming-mice ranged from ~10^2.0^ to 10^3.5^ TCID_50_ (Table 1).

### 3.3. Phylogenetic Analysis

PRV *gE* and *gC* of the newly identified 15 PRV stains were amplified by PCR and cloned into a pUCm-T vector for sequencing [2]. According to the phylogenetic analysis based on PRV *gE* or *gC* sequences, all PRV strains, including 15 novel PRV strains and reference strains, were divided into two genotypes: genotype I and genotype II (Figure 1A,B). In agreement with a previous study [14], most of the isolates from China were clustered as genotype II, which could be further divided into the classical (before 2012) and variant (after 2012) sub-genotypes, while PRV strains from other parts, such as Europe and the U.S., belonged to genotype I. Notably, all 15 PRV isolates obtained in this study belonged to genotype II. Importantly, the *gE* phylogenetic tree showed that one isolate from Yunnan Province in 2017 (designed as YuN-YL-2017) was genetically closer to classical PRV strains compared with others (Figure 1A), while the *gC* gene was in the same clade with other PRV variants (Figure 1B).

### 3.4. Analysis of PRV gC and gE

The nucleotide and the corresponding amino acid sequence variations for *gC* (1464 bp) and *gE* (1734~1740 bp) genes of 15 novel PRV strains within the isolates were 0.0~0.3%, 0.0~0.7%, and 0.0~0.8%, 0.0~1.7%, respectively (Table 4). Moreover, compared with PRV variants and classical PRV strains, these 15 PRV strains exhibited a 99.6~100.0%, 99.1~99.4% nucleotide and 99.4~100.0%, 98.3~98.7% amino acid sequence identity in the gC gene and a 99.3~100.0%, 99.2~99.8% nucleotide and 98.6~100.0%, 98.8~99.5% amino acid sequence identity in the gE gene (Table 4), respectively. Remarkable, the *gE* gene of the YuN-YL-2017 strain showed a higher sequence homology with classical PRV strains (such as Ea and Fa), while its *gC* gene was highly homologous to the variants (such as HeN1 and ZJ01).

PRV gC and gE proteins sequences among the PRV strains were further aligned. The results revealed that there was no amino acid insertions or deletions, but several mutations were observed among gC proteins of 15 PRV strains when compared with other PRV variants. Except for some amino acid mutations among gE proteins, compared with the PRV variants, the YuN-YL-2017 strain had two amino acid deletions at site 48 (D) and 498 (D).

## 4. Discussion

Since the emergence of variant PRV strains in China in 2011, the disease caused by PRV variants has been considered a major factor contributing to huge economic losses to the swine industry. Recently, the cross-species transmission events of PRV from pigs to humans have also attracted increasing attention [15,16]. Great efforts have been made for the control of PR; particularly, this disease was listed in the “Mid- and Long-term Animal Disease Prevention and Control Program in China (2012–2020)”. Nevertheless, PR remains widely spread in Chinese pig populations and pose a challenge for other animals breeding in China, such as fox and mink. Thus, obtaining accurate data on the epidemiological characteristics of PRV is beneficial for formulating control or eradication measures.

The present results showed that the average PRV-gE seropositive rate was 31.37% among 20,158 serum samples from Yunnan Province from 2017 to 2021. Further analysis showed that the PRV seroprevalence in Yunnan Province between September 2018 to January 2020 (41.48%, 2449/5904) was higher than these during March 2017–August 2018 (29.25%, 2355/8051) and April 2020–December 2021 (24.50%, 1520/6203), and a similar epidemiological trend was also observed in the pathogen detection section in this study. Since the outbreaks of African swine fever (ASF) and its rapid spread since August 2018 contributed to the substantial reduction of the sow population in China, many PRV-positive sows might have been introduced into pig farms to keep the breeding scale, which contributed to the high seroprevalence of PRV in some regions of China [2]. Owing to the fact that the prevalence of ASF has been controlled in 2020 [17] and the excessive pig production in China recently, many pig farms subsequently focused on the prevention or eradication of other infectious diseases, including PR, classical swine fever, etc.

Two factors, “pig herd” and “breeding scale”, were significantly associated with the seroprevalence of PRV of pigs in Yunnan Province. The seroprevalence of PRV in fattening pigs, sows, and gilts was higher than these in piglets, nursery pigs, and boars; similar results were also observed in previous research [18]. On one hand, the occurrences of PR in fattening pigs are often neglected since they only display mild symptoms. On the other hand, fattening pigs are not immunized with PRV vaccines in some pig farms. Meanwhile, long-term feeding increases the probability of PRV infection among sows and gilts. Moreover, as reported in Lin’s study [2], we also found that a lower PRV-gE seropositive rate among pigs was detected from medium-sized farms compared with those in large and small ones, which might suggest that the medium-density feeding mode is more suitable for infectious diseases control.

The gC protein participates in viral abortion on a host cellular surface; meanwhile, this protein is an important target for neutralizing an antibody [19]. The gE protein is mainly involved in viral virulence [4]. Phylogenetic analysis based on the *gE* or *gC* gene revealed that PRV strains prevalent worldwide can be divided into two genotypes (namely, genotype I and genotype II), and most PRV strains circulated in China belong to the genotype II [1,14]. In line with these, 15 novel PRV strains obtained in this study formed one large clade with Chinese PRV variants (after 2012) and Chinese classical PRV strains (before 2012) and belonged to the genotype II, which showed a distinct relationship to genotype I strains, such as Bartha and Backer (Figure 1A,B). Remarkably, one isolate, namely, YuN-YL-2017, was identified as a PRV variant according to the genetic analysis of *gC* gene, which belonged to the classical strains according to the *gE* gene. These results indicated this strain might be a recombinant variant strain. Further analysis showed that the LD_50_ of YuN-YL-2017 to mice was higher than those of other PRV variants (10^3.5^ TCID_50_ VS 10^2.0–2.8^ TCID_50_), suggesting that the recombinant event in the genome of YuN-YL-2017 decreased its virulence to mice, and the underlying mechanisms will be explored in the future.

## 5. Conclusions

In conclusion, this study comprehensively investigated the prevalence and genetic features of PRV in Yunnan Province from 2017 to 2021, showing that PR remains highly prevalent among pig populations in Yunnan Province, China. Phylogenetic analysis showed that all 15 PRV strains isolated in this study belonged to the genotype II, displaying a distinct evolutionary relationship with the Bartha strain in genotype I, which might partly explain the immune failure of the PRV Bartha-K61 vaccine in pigs challenged by PRV variants, and further suggesting that novel vaccines should be developed for the control of PR in this region. In addition, the results above also highlighted the importance of continuous monitoring the molecular epidemiology of such recombinant PRV strains in the future.

## Figures and Tables

**Figure 1 viruses-14-00895-f001:**
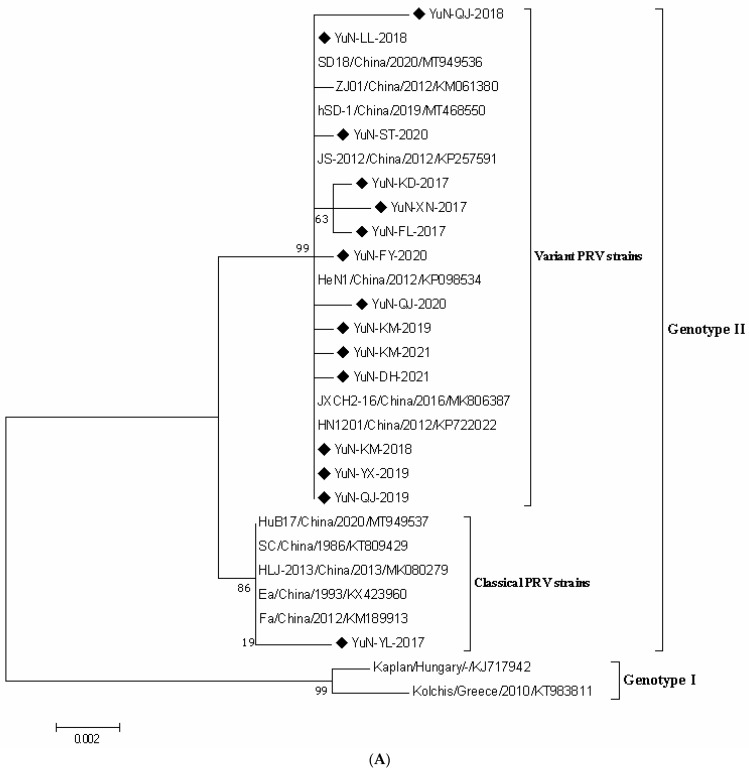
Phylogenetic analysis based on the nucleotide sequences of *gE* (**A**) and *gC* (**B**) genes of the 15 novel PRV isolates obtained in this study and other reference strains. A phylogenetic tree was generated using the neighbor-joining method with 1000 bootstrap replicates in MEGA X software. The black triangle represents the 15 PRV isolates.

**Table 1 viruses-14-00895-t001:** Detailed information of PRV strains identified in this study and reference strains, including strain name, collection year, isolation region, viral titer, the median lethal doses (LD_50_) to mice, and GenBank accession numbers.

Strains	Collection Year	Isolation Region	Pig Farm Size	Tissue Type	TCID_50_/0.1 mL	LD_50_	GenBank Accession
YuN-YL-2017	2017	Yunan, China	Small	Lung, fattening pig	10^5.25^	10^3.5^	OM982597(*gC*), ON012780 (*gE*)
YuN-KD-2017	2017	Yunan, China	Large	Aborted fetus	10^6.58^	10^2.65^	OM982598 (*gC*), ON012781 (*gE*)
YuN-XN-2017	2017	Yunan, China	Large	Aborted fetus	10^5.75^	10^2.85^	OM982599 (*gC*), ON012782 (*gE*)
YuN-FL-2017	2017	Yunan, China	Medium	Aborted fetus	10^6.083^	10^2.63^	OM982600 (*gC*), ON012783 (*gE*)
YuN-QJ-2018	2018	Yunan, China	Medium	Aborted fetus	10^6.5^	10^2.5^	OM982601 (*gC*), ON012784 (*gE*)
YuN-LL-2018	2018	Yunan, China	Large	Aborted fetus	10^6.875^	10^2.08^	OM982602 (*gC*), ON012785 (*gE*)
YuN-KM-2018	2018	Yunan, China	Small	Aborted fetus	10^7.0^	10^2.85^	OM982603 (*gC*), ON012786 (*gE*)
YuN-YX-2019	2019	Yunan, China	Medium	Aborted fetus	10^6.0^	10^1.80^	OM982604 (*gC*), ON012787 (*gE*)
YuN-KM-2019	2019	Yunan, China	Large	Aborted fetus	10^6.38^	10^2.0^	OM982605 (*gC*), ON012788 (*gE*)
YuN-QJ-2019	2019	Yunan, China	Small	Aborted fetus	10^6.59^	10^2.5^	OM982606 (*gC*), ON012789 (*gE*)
YuN-FY-2020	2020	Yunan, China	Large	Aborted fetus	10^7.12^	10^2.43^	OM982607 (*gC*), ON012790 (*gE*)
YuN-QJ-2020	2020	Yunan, China	Small	Aborted fetus	10^6.0^	10^2.63^	OM982608 (*gC*), ON012791 (*gE*)
YuN-ST-2020	2020	Yunan, China	Large	Aborted fetus	10^7.0^	10^2.5^	OM982609 (*gC*), ON012792 (*gE*)
YuN-DH-2021	2021	Yunan, China	Medium	Aborted fetus	10^6.67^	10^2.38^	OM982610 (*gC*), ON012793 (*gE*)
YuN-KM-2021	2021	Yunan, China	Large	Aborted fetus	10^6.25^	10^2.43^	OM982611 (*gC*), ON012794 (*gE*)
hSD-1	2019	Shandong, China			-	-	MT468550
JXCH2-16	2016	Jiangxi, China			-	-	MK806387
SD-18	2020	China			-	-	MT949536
HN1201	2012	Henan, China			-	-	KP722022
ZJ01	2012	Zhejiang, China			-	-	KM061380
SC	1986	Sichuan, China			-	-	KT809429
HLJ-2013	2013	Heilongjiang, China			-	-	MK080279
HeN1	2012	Henan, China			-	-	KP098534
Ea	1993	Hubei, China			-	-	KX423960
HuB17	2020	Hubei, China			-	-	MT949537
Fa	2012	Fujian, China			-	-	KM189913
JS-2012	2012	Jiangsu, China			-	-	KP257591
Bartha	-	Hungary			-	-	JF797217
Kaplan	-	Hungary			-	-	KJ717942
Kolchis	2010	Greece			-	-	KT983811

**Table 2 viruses-14-00895-t002:** Seroprevalence of PRV-gE among pigs in Yunnan province with different risk factors.

	Category	No. Sample	No. Positive	% (95% CI)	*p*-Value
Period	March 2017 to August 2018	8051	2355	29.25 (28.26–30.24)	<0.001
	September 2018 to January 2020	5904	2449	41.48 (40.22–42.74)	<0.001
	April 2020 to December 2021	6203	1520	24.50 (23.43–25.57)	Reference
Pig herd	Piglets	2644	442	16.72 (15.29–18.14)	Reference
	Nursery pigs	5304	1467	27.66 (26.45–28.86)	<0.001
	Fattening pigs	5621	2301	40.94 (39.65–42.22)	<0.001
	Sows	4039	1245	30.82 (29.40–32.24)	<0.001
	Gilts	2123	761	35.84 (33.81–37.89)	<0.001
	Boars	427	108	25.29 (21.17–29.42)	<0.001
Pig farm size	Small	3438	1273	37.03 (35.41–39.64)	<0.001
	Medium	6273	1556	24.80 (23.74–25.87)	Reference
	Large	10,447	3495	33.45 (32.55–34.36)	<0.001
		20,158	6324	31.37 (30.73–32.01)	

**Table 3 viruses-14-00895-t003:** The PRV-gE DNA positive rates among pigs with different risk factors.

	Category	No. Sample	No. Positive	% (95% CI)	*p*-Value
Period	March 2017 to August 2018	177	16	9.04 (4.82–13.26)	0.947
	September 2018 to January 2020	103	15	14.56 (7.75–21.38)	0.165
	April 2020 to December 2021	136	12	8.82 (4.06–13.59)	Reference
Samples	Aborted fetus	108	15	13.89 (7.37–20.41)	< 0.01
	Piglets with neurological symptoms	83	15	18.07 (9.79–26.35)	< 0.01
	Others	225	13	5.78 (2.73–8.83)	Reference
		416	43	10.33 (7.41–13.26)	

**Table 4 viruses-14-00895-t004:** Sequence similarity analysis of the gC and gE sequences of PRV strains identified in this study.

Selected Strains	Nucleotide Sequences (%)		Amino Acid Sequences (%)	
	*gC*	*gE*	gC	gE
15 PRV strains obtained in this study	99.7~100.0	99.3~100.0	99.2~100.0	98.3~100.0
Compared with PRV variants	99.6~100.0	99.3~100.0	99.4~100.0	98.6~100.0
Compared with classical PRV strains	99.1~99.4	99.2~99.8	98.3~98.7	98.8~99.5
Compared with PRV strains in genotype I	94.2~96.1	97.4~97.8	89.2~96.5	95.3~96.0

## Data Availability

All data presented in the present study can be found in online repositories.

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
