# Peer review of "Epidemiological Investigation and Genetic Analysis of Pseudorabies Virus in Yunnan Province of China from 2017 to 2021"

_viruses, 2022, doi:10.3390/v14050895_

Round 1

Reviewer 1 Report

Pseudorabies is a important disease in swine production and it can infect humans. The authors of this manuscript detailed important information about risk factors in China's farms also it raises concerning for human health. The article is well written and data is well described. 

Minor: 

Please add more information about differences in genotype I and Genotype II in the introduction.

Author Response

Pseudorabies is a important disease in swine production and it can infect humans. The authors of this manuscript detailed important information about risk factors in China's farms also it raises concerning for human health. The article is well written and data is well described. 

Answer: The authors do appreciate the reviewer for evaluating our manuscript and recognizing its merits. We have responded the comments in a point-to-point way, which are indicated in the following.

Minor: 

Please add more information about differences in genotype I and Genotype II in the introduction.

Answer: Thanks for your suggestion, the genetic characteristics among PRV genotype I and genotype II strains has been clarified in the introduction.

Reviewer 2 Report

Dear Editor,

The manuscript entitled “Epidemiological investigation and genetic analysis of pseudorabies virus in Yunnan Province of China from 2017 to 2021” by Yao et al. describes the serological and epidemiological investigations of PRV in pig populations from Yunnan Province from 2017 to 2021. The results showed that 37.37% of serum samples were positive for PRV glycoprotein E (gE)-specific antibodies via enzyme-linked immunosorbent assay (ELISA). Of the 416 tissue samples collected from PRV-suspected pigs in Yunnan Province, 43 samples were positive for PRV nucleic acid. 15 novel PRV strains from these PRV-positive samples were isolated, whose gC and gE sequences were analyzed. Phylogenetic analysis showed that all 15 isolates obtained in this study belonged to genotype II. Additionally, the gC gene of one isolate (YuN-YL-2017) was genetically closer to variant PRV strain than others, while gE of which was in the same clade as other classical PRV strains, indicating that this isolate might be a recombinant strain generated from the classical and variants strains. The results of this study, evidenced the severe PRV epidemic in Yunnan Province, and indicated that PRV variants is the major genotypes threatening pig industry development.

I think this manuscript can be worth publishing if the following points are inserted:

General comments

1) Please, indicate if an Ethics Committee has authorized the experiments;

2) Please, in Materials and Methods section (Samples collection), transfer the “573 pig farms were included in this survey where nearly all sampled pig had been immunized with attenuated PRVvaccine (Bartha-K61 or HB-98 strain) or inactivated PRV vaccine” part gives in section 3.1;

3) Please, in Results section (Phylogenetic analysis), transfer the “PRV gE and gC of the newly identified 15 PRV strain were amplified by PCR, and cloned into pUCm-T vector for sequencing [2](Lin et al; 2021) part gives in section 3.3;

4) Please, describe the farms where the viruses were isolated and the type of sample used for viral isolation.

5) Please, indicate the titres of the viruses isolated.

Author Response

The manuscript entitled “Epidemiological investigation and genetic analysis of pseudorabies virus in Yunnan Province of China from 2017 to 2021” by Yao et al. describes the serological and epidemiological investigations of PRV in pig populations from Yunnan Province from 2017 to 2021. The results showed that 37.37% of serum samples were positive for PRV glycoprotein E (gE)-specific antibodies via enzyme-linked immunosorbent assay (ELISA). Of the 416 tissue samples collected from PRV-suspected pigs in Yunnan Province, 43 samples were positive for PRV nucleic acid. 15 novel PRV strains from these PRV-positive samples were isolated, whose gC and gE sequences were analyzed. Phylogenetic analysis showed that all 15 isolates obtained in this study belonged to genotype II. Additionally, the gC gene of one isolate (YuN-YL-2017) was genetically closer to variant PRV strain than others, while gE of which was in the same clade as other classical PRV strains, indicating that this isolate might be a recombinant strain generated from the classical and variants strains. The results of this study, evidenced the severe PRV epidemic in Yunnan Province, and indicated that PRV variants is the major genotypes threatening pig industry development.

 Answer: The authors do appreciate the reviewer for evaluating our manuscript and appreciating its merits. We have responded the comments in a point-to-point way, which are indicated in the following.

I think this manuscript can be worth publishing if the following points are inserted:

General comments

  • Please, indicate if an Ethics Committee has authorized the experiments;

Answer: thanks for your consideration, the ethics committee has authorized the experiments, and the information of the ethics committee has been provided in this article.

  • Please, in Materials and Methods section (Samples collection), transfer the “573 pig farms were included in this survey where nearly all sampled pig had been immunized with attenuated PRVvaccine (Bartha-K61 or HB-98 strain) or inactivated PRV vaccine” part gives in section 3.1;

Answer: Thanks for your suggestion, we have noticed that the information of sample collection has been transferred in the section 3.1, and we totally agree with it.

  • Please, in Results section (Phylogenetic analysis), transfer the “PRV gE and gC of the newly identified 15 PRV strain were amplified by PCR, and cloned into pUCm-T vector for sequencing [2](Lin et al; 2021) part gives in section 3.3;

Answer: Thanks for your suggestion, we have noticed that the information of sample collection has been transferred in the section 3.1, and we totally agree with it. And we have deleted the words “Lin et al., 2021”.

  • Please, describe the farms where the viruses were isolated and the type of sample used for viral isolation.

Answer: thanks for your suggestion, here we have described the pig farm size where the viruses were isolated, which was shown in the Table 1, and nearly all viruses were isolated from the aborted fetus expect for YuN-YL-2017, as shown in the Table 1.

  • Please, indicate the titres of the viruses isolated.

Answer: thanks for your consideration here, we have indicated the viral titres of the isolated PRV strains in the Table 1, and simply described it in the results section 3.3.
